# Determinants of birth registration in India: Evidence from NFHS 2015–16

Krishna Kumar [ORCID]*⍟, Nandita Saikia [ORCID]⍟

Centre for the Study of Regional Development, School of Social Sciences III, Jawaharlal Nehru University, New Delhi, India

⍟ These authors contributed equally to this work.
* krishhna94@gmail.com

**Data Availability Statement:** The data underlying the results presented in the study are available from https://dhsprogram.com/Data/.

**Funding:** The authors received no specific funding for this work.

## Abstract

### Objectives

Official data on birth is important to monitor the specific targets of SDGs. About 2.7 million children under age five years do not have official birth registration document in India. Unavailability of birth registration document may deprive the children from access to government-aided essential services such as fixed years of formal education, healthcare, and legal protection. This study examines the effect of socioeconomic, demographic and health care factors on birth registration in India. We also examined the spatial pattern of completeness of birth registration that could be useful for district level intervention.

### Methods

We used data from the National Family Health Survey (NFHS-4), 2015–16. We carried out the descriptive statistics and bivariate analysis. Besides, we used multilevel binary logistic regression to identify significant covariates of birth registration at the individual, district, and state levels. We used GIS software to do spatial mapping of completeness of birth registration at district level.

### Results

The birth registration level was lower than national average (80.21%) in the 254 districts. In Uttar Pradesh, 12 out of 71 districts recorded lower than 50% birth registration. Also, some districts from Arunachal Pradesh, J&K, and Rajasthan recorded lower than 50% birth registration. We also found a lower proportion of children are registered among children of birth order three and above (62.83%) and rural resident (76.62%). Children of mothers with no formal education, no media exposure, poorest wealth quintile, OBC and muslims religion have lower level of birth registration. Multilevel regression result showed 25 percent variation in birth registration lie between states while the remaining 75 percent variation lie within states. Moreover, children among illiterate mother (AOR = 0.57, CI [0.54, 0.61], p<0.001), Muslims households (AOR = 0.90, CI [0.87, 0.94], p<0.001), and poorest wealth quintile (AOR = 0.38, CI [0.36, 0.41], p<0.001) showed lower odds for child's birth registration.

**Competing interests:** The authors have declared that no competing interests exist.

## Conclusion

We strongly suggest linking the birth registration facilities with health institutions.

## Introduction

Birth registration is the formal recording of the occurrence and characteristics of birth by the civil registrar with legal requirements. UNICEF documented "the child should be registered immediately after birth and shall have the right from birth to a name and right to acquire nationality" [1]. The convention on rights of the child also recognised the right of every child to birth registration [2]. A birth certificate documents essential information such as age, place of birth, and family background [3]. Besides being official documentation of a child's birth, it facilitates access to government-provided essential services such as education, health facilities, and legal protection [4–7]. It is found that illegal practices such as child labour and trafficking are negatively associated with children's birth registration [8, 9]. It is challenging for a child to claim legal protection without official documentation of his/her birth. Birth registration may affect the survival and holistic development of a child [10]. Previous studies also showed that where children have not provided with a citizenship right through legal documentation of their birth, the right of individual to access to civic, political and social identities are compromised [2, 11].

Moreover, the quality of vital statistics is indispensible to monitor children's development. Timely birth registration is essential for generating an up-to-date and reliable vital statistics [10]. Complete data on birth registration is crucial for policymakers and health officials for studying fertility patterns at the national and sub-national levels. Sustainable Development Goals (SDGs) include a dedicated target under Goal 16: "the aim of providing legal identity for all, including birth registration, by 2030" [12]. It is essential to generate complete and timely birth statistics for monitoring and tracking the progress towards SDGs.

Despite government agencies and UNICEF's effort to universalize birth registration globally, about 166 million children under age five and 40 million infants were not officially documented [3]. Moreover, receiving a birth certificate is particularly challenging in parts of Africa and Asia. A high proportion of children under age five years was not registered in South Asia. According to the UNICEF's report, about 77 million children under age five do not have a birth certificate in South Asia. There has been a large disparity among countries in terms of birth registration. High-income countries like United States, United Kingdom, Australia, and Germany recorded 100 percent of birth [3], and issued birth certificates to the most children [3]. On the other hand, low or middle-income countries showed more unsatisfactory performance in registering the child's birth [3].

In India, the Births, Deaths and Marriage Registration Act, was enacted in 1886, suggested voluntary registration of births and deaths. However, it was not adequately implemented across India. After independence, India's Government introduced the Registration of Births and Deaths Act in 1969, which mandates registration of all births and deaths within 21 days [13]. Despite the provision of mandatory birth registration, nearly 20 percent of children under age five years were not registered, and one-fifths of registered children do not have a birth certificate [14]. Also, about 2.7 million children under age five are not registered in India in 2020. However, there has been increased birth registration levels from 76% in 2008 to 89% in 2018 [13]. Further, there has been enormous disparity at the state and district level in terms of coverage and access to birth registration in India [13].

There has been increased research around the impact of under-registration, but previous studies were focused on the need and benefits associated with functional registration systems [4, 6, 7, 12]. Previous studies showed institutional birth, mother's health seeking behaviour, parents education, caste, religion and wealth status are significant determinants of birth registration [15, 16]. A few studies attempted to investigate the effect of maternal autonomy and ability, and utilisation of perinatal health services on child's birth registration [15, 17]. Another study showed Civil Registration System's design and functional status [18]. However, previous studies is limited to small sample size, based on few districts of India and mostly in the context of other countries. As of our knowledge, there is a lack of systematic research examining predictors of birth registration in India at an individual, district and state level. This study investigates demographic, socioeconomic, and healthcare predictors associated with birth registration in India. We also present the spatial pattern of completeness of birth registration that could be useful for district-level intervention.

## Materials and methods

### Data source

We used data from National Family Health Survey, 2015–16 (NFHS-4). It provides essential information on household populations, housing characteristics, basic demographic and socioeconomic characteristics of respondents, fertility, family planning, maternal and child health, infant and child mortality, nutrition, morbidity including adult health issues, women empowerment, and domestic violence at the nation, state and district level. This survey was conducted under the Ministry of Health and Family Welfare (MoHFW) leadership and managed by the International Institute of Population Sciences (IIPS), Mumbai. The NFHS provides information on the number of dejure children under age five registered by the civil registrar. In the survey, a question on birth registration was asked as "*Does a child have a birth certificate or has child's birth ever been registered by the civil authority*" [14].

### Study design and samples

Two stages stratified random sampling approach was adopted in this survey. Primary Sampling Units (PSUs) (villages in rural areas and census enumeration blocks in urban areas) are selected using probability proportional to population size at the first stage. Subsequently, an equal number of households were selected from each PSU through systematic random sampling. In total, 6,99,686 women and 1,12,122 men were interviewed in this survey. We included a total of 2,25,867 children under age five years from 640 districts and 36 states/UTs of India in the final analysis sample.

### Outcome variable

A dependent variable birth registered was defined as one equals to children under age five years who have a birth certificate or ever been registered by the civil authority, otherwise 0.

### Predictor variables

We considered demographic, socioeconomic, and healthcare characteristics to identify factors associated with children's birth registration. We categorized child age as 0–1,1–3, and 3–5 years, sex of the child (male or female), and birth order as 1,2,3 and 3+. Other demographic and socioeconomic characteristics include the place of residence (urban or rural), sex of the head of household (male or female), mothers age in years (15–24, 25–34, 35–49), mother's level of education (illiterate, primary, secondary and higher), religion (Hindus, Muslims and

others), caste (Scheduled Castes/S.Cs., Scheduled Tribes/S.Ts., Other Backward Class/OBC, and others), wealth quintile (poorest, poorer, middle, richer and richest). We categorized marital status into two categories (currently married and separated/divorced/widow). We defined mothers' exposure to media into three categories (no, partial and full) based on their response to how often they read the newspaper, listen to the radio, and watch television. Mothers who did not read newspaper, not listen to radio and not watch television less than or at least once in a week were categorised as having no media exposure. Mothers exposed to any one or two of the three forms of media were categorised as having partial media exposure. Mothers exposed to all the three forms of media were categorised as having full media exposure. Besides we included a healthcare variable. We divided children's vaccination status into three categories (no, partial and full). No vaccination refers to children aged 12–23 months who did not receive any vaccines since birth, partial vaccination indicates children received at least one but not all recommended vaccines and full vaccination refers who received all 13 recommended vaccines.

Furthermore, we considered district-level factors such as the proportion of SCs, the proportion of children (12–23 months) receiving full immunization, and the proportion of institutional birth. We generated the district level variables by aggregating individual or household level information at the district level.

## Statistical analyses

We presented the descriptive statistics of dependent and independent variables included in this study. Further, we analyzed bivariate distribution to examine the association of demographic, socioeconomic and health care variables with children's birth registration. Also, we performed chi-square test to identify the significance of such associations. We applied multilevel binary logistic regression models with random intercept and fixed slope to calculate odds ratio (OR)/Adjusted odds ratio (AOR) at three levels (level 1: Individual; level 2: district; level 3: state) with 95 percent of confidence interval (CI) and p-value. When the p-value was lesser than 0.05, odds ratios were considered statistically significant. Multilevel analysis generates variance at each level, providing the technical advantage of assessing unobserved effects at each level. The hierarchical model of the survey justified the application of multilevel modelling in this study. We fitted four models. Firstly, we run the null model. Second model included only demographic variables whereas the third model included demographic and socioeconomic variables. Finally, the fourth model was adjusted for demographic, socioeconomic, and district-level variables. We used Akaike Information Criteria (AIC) and log-likelihood for model comparison. The model with the lowest value of AIC and the highest log-likelihood value was considered the best fit. Besides, we checked multicollinearity using the Variance Inflation Factor (VIF). We found no evidence of collinearity among the included independent variables (mean VIF = 1.34). We explained the fourth model in detail as there was a similar pattern in the second, third and fourth models. We also estimated Intra Class Correlation (ICC) to find the percentage variance explained at district and state level. All analysis was performed using R (version 4.0.2). Further, we also mapped the district-wise proportion of registered children using the Geographic Information System (GIS). Besides, we mapped the predicted estimates of birth registration level using the GIS software.

The mathematical equation of the three-level model is shown below:

$$logit(\pi ijk) = \log(\pi ijk/(1 - \pi ijk)$$
$$= \beta 0jk + \beta 1x1ijk + \beta 2x2ijk + \cdots\cdots\cdots \beta nxijk + u0jk + v0jk + eijk$$

Where $\pi ijk$ = p(Yijk = 1) is the probability of a child i in the district j, from state k,

registered birth. Yijk would equal one if a child were registered, otherwise 0. The probability is defined as a function of an intercept and the explanatory variables. $\beta0jk = \beta0 + \mu0jk$, where $\beta0jk$ shows that intercept was random at jth (district) and kth (state) levels. The variables $X1ijk$ to $Xnijk$ were exploratory variables and their corresponding regression coefficients ($\beta1,\beta2,\ldots \beta n$) were fixed effects.

u0jk is the random state effect assumed to be normally distributed with $N(0,\sigma u^2)$

v0jk is the random district effect assumed to be normally distributed with $N(0,\sigma v^2)$

eijk is the random errors assumed to be normal with $N(0, \sigma e^2)$ and independent of random effects at level 2 and level 3.

## Results

Fig 1 shows the level of birth registration of children under age five years who have ever been registered by districts of India. We found lower birth registration was recorded in Uttar Pradesh, Bihar, Arunachal Pradesh and Rajasthan. In Uttar Pradesh, 12 out of 71 districts recorded lower than 50 per cent birth registration. Besides, four Arunachal Pradesh districts, Purba Champaran in Bihar, Rajouri in J&K, and Dhaulpur in Rajasthan, recorded lower than 50 percent birth registration. Shahjahanpur (23.54%), Tawang (29.86%), Balrampur (31.53%) were the worst-performing districts regarding the level of birth registration. On the other hand, 54 districts of India recorded birth registration level above 99 percent. Gurudaspur (100%) and Faridkot (100%) district of Punjab and six district of Tamil Nadu, north and south district of Delhi recorded 100 percent birth registration. Fig 2 shows predicted estimates of birth registration level by districts of India. Predicted birth registration estimates showed 11 districts of Uttar Pradesh, four Arunachal Pradesh districts, Purba champaran in Bihar, Rajouri in J&K and Dhaulpur in Rajasthan recorded lower than 50 percent birth registration level. We found there was marginal difference between observed and predicted estimates of birth registration level (Figs 1 and 2).

Table 1 shows the descriptive statistics of dependent and independent variables included in the study. We found 62.7 percent of children in our sample have birth certificates. Besides, 17.5 percent of children was registered with the civil authorities, however, they did not have a certificate. Around 52 percent children of our sample are male and 29 percent children are urban residents. Around 19 percent, 40 percent and 41 percent children belonged to age group 0–1, 1–3 and 3–5 years respectively. Nearly 38 percent and 32 percent children belonged to birth order 1 and 2 respectively.

Around 57 percent mothers belonged to age group 25–34 years and 10 percent mothers had received higher education. A 99 percent mothers of our sample are currently married and 66 percent mothers have partial media exposure. Moreover, 88 percent households included in the study are male headed households. Hindus are 79 percent of our sample. About 23 percent and 46 percent households belonged to SCs and OBC respectively. Further, 25 percent, 20 percent and 15 percent households belonged to poorest, middle and richest wealth quintile respectively. Also, around 51 percent of children of our sample are fully immunised.

Table 2 shows the proportion of children under age five whose birth has ever been registered by baseline characteristics. Result shows a marginal difference in birth registration by sex of the child (male-79.91%, female-80.53%). Birth registration was the highest among children aged between 1 to 3 years (81.75%). Moreover, a low proportion of children among birth order of three and above (62.83%) are found to be registered compared with children among birth order 1 (86.91%) and 2 (82.97%). There is a significant association between place of residence and birth registration, showing a lower proportion of birth registration in rural areas than urban areas (76.62% vs. 89.14%). We found child's vaccination status positively affects birth

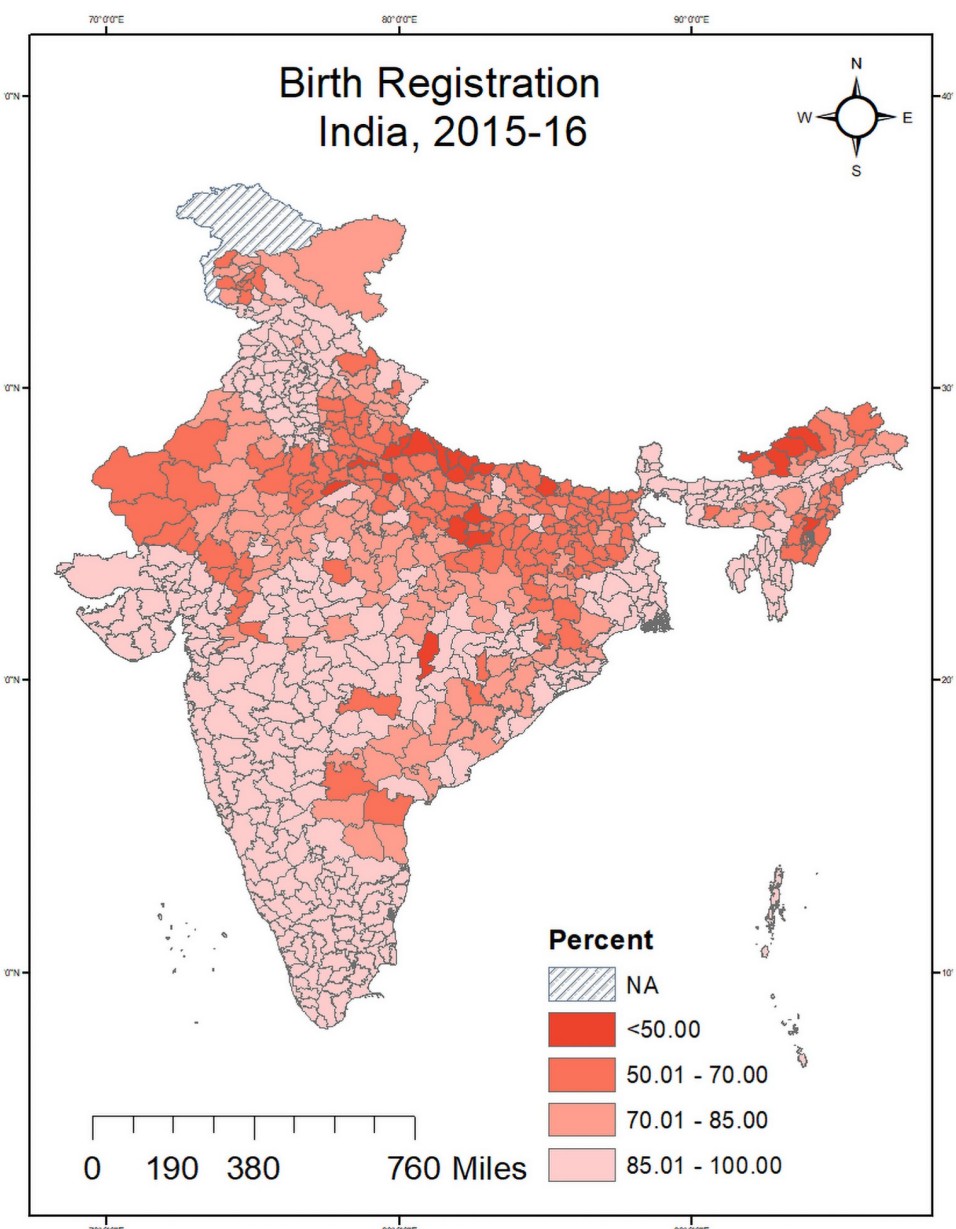

**Fig 1. Level of birth registration of children under age five years, Indian districts, 2016.** Source—Author generated the map using GIS.

registration level. Proportion of registered children is higher among fully vaccinated children (85.70%) as compared to children who received no vaccination (61.70%). Besides, a higher proportion of children (81.75%) was registered among mothers aged 25–34 years. Also, we found the lower practice of child's birth registration among illiterate mother (64.35%) compared to higher educated mother (91.95%). We found that among 99 percent currently married mothers of our sample 80.24% of children are registered. However, result was not significant. The child's birth registration practice was the lowest among mothers who had no media exposure (64.62%).

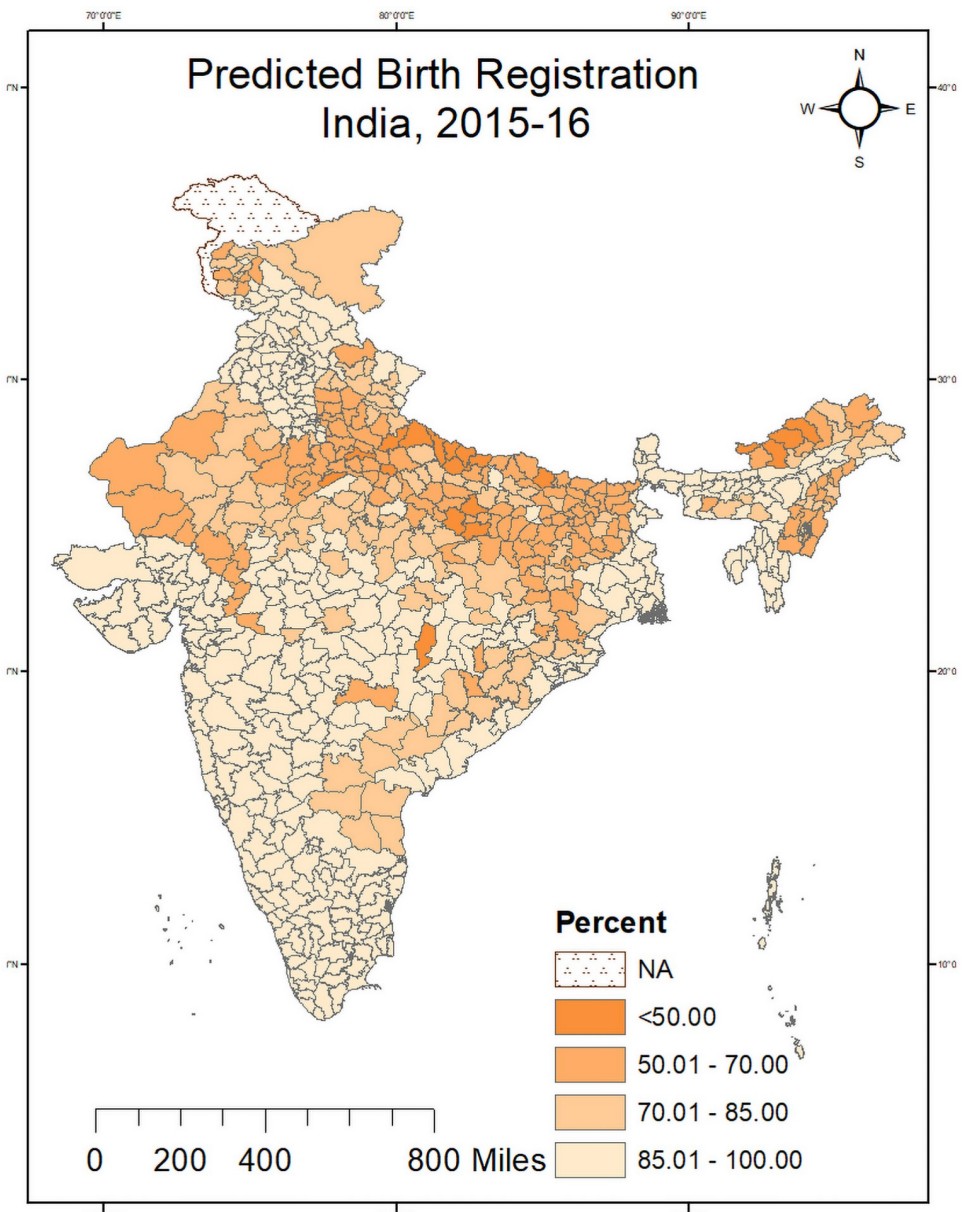

**Fig 2. Predicted estimates of birth registration level of children under age five years, Indian districts, 2016.**
Source- Author generated the map using GIS.

Moreover, we found that household characteristics are significantly associated with birth registration. This study also shows that a marginally lower proportion of children was registered in Muslims households (77.87%) than Hindus households (80.15%). Besides, a lower percentage of children was registered among STs (76%) and OBC (78.09%). Nearly 94.44 percent of children are registered among the richest household, whereas about 64.32 percent of children are registered among the poorest household. It is also found that the proportion of registered children were lower among female-headed household (77.20%).

Table 3 shows the result of multilevel binary logistic regression of demographic, socioeconomic and health care factors. We showed AOR, CI and p-value of explanatory variables

**Table 1. Descriptive statistics.**

| | Proportion | Std. Error | 95% confidence interval | |
|---|---|---|---|---|
| Descriptive statistics (weighted sample size = 218635) | | | | |
| | | | Lower | Upper |
| **Birth registration** | | | | |
| Not birth registered | 0.198 | 0.001 | 0.183 | 0.206 |
| Have a birth certificate | 0.627 | 0.001 | 0.625 | 0.629 |
| Registered but not have a certificate | 0.175 | 0.001 | 0.174 | 0.177 |
| **Sex of child** | | | | |
| Male | 0.522 | 0.001 | 0.520 | 0.524 |
| Female | 0.478 | 0.001 | 0.476 | 0.480 |
| **Child's age (in year)** | | | | |
| 0–1 | 0.191 | 0.001 | 0.190 | 0.193 |
| 1–3 | 0.400 | 0.001 | 0.398 | 0.402 |
| 3–5 | 0.409 | 0.001 | 0.407 | 0.411 |
| **Birth Order** | | | | |
| 1 | 0.375 | 0.001 | 0.373 | 0.377 |
| 2 | 0.324 | 0.001 | 0.322 | 0.326 |
| 3 | 0.155 | 0.001 | 0.154 | 0.157 |
| 3+ | 0.145 | 0.001 | 0.143 | 0.146 |
| **Place of residence** | | | | |
| Urban | 0.287 | 0.001 | 0.285 | 0.289 |
| Rural | 0.713 | 0.001 | 0.711 | 0.715 |
| **Mother's age** | | | | |
| 15–24 | 0.342 | 0.001 | 0.340 | 0.344 |
| 25–34 | 0.571 | 0.001 | 0.569 | 0.573 |
| 35–49 | 0.086 | 0.001 | 0.085 | 0.088 |
| **Mother's education** | | | | |
| Illiterate | 0.301 | 0.001 | 0.103 | 0.105 |
| Primary | 0.139 | 0.001 | 0.453 | 0.457 |
| Secondary | 0.455 | 0.001 | 0.138 | 0.141 |
| Higher | 0.104 | 0.001 | 0.299 | 0.303 |
| **Marital Status** | | | | |
| Currently married | 0.989 | 0.002 | 0.988 | 0.989 |
| Separated/Divorced/widowed | 0.011 | 0.001 | 0.011 | 0.012 |
| **Media exposure** | | | | |
| No | 0.269 | 0.001 | 0.267 | 0.271 |
| Partial | 0.660 | 0.001 | 0.658 | 0.662 |
| All | 0.071 | 0.001 | 0.069 | 0.072 |
| **Sex of the head of household** | | | | |
| Male | 0.881 | 0.001 | 0.880 | 0.882 |
| Female | 0.119 | 0.001 | 0.118 | 0.120 |
| **Religion** | | | | |
| Hindus | 0.785 | 0.001 | 0.783 | 0.786 |
| Muslims | 0.166 | 0.001 | 0.165 | 0.168 |
| Others | 0.049 | 0.002 | 0.048 | 0.050 |
| **Caste** | | | | |
| SCs | 0.226 | 0.001 | 0.224 | 0.227 |
| STs | 0.111 | 0.001 | 0.110 | 0.113 |

(*Continued*)

**Table 1.** (Continued)

| Descriptive statistics (weighted sample size = 218635) | | | | |
|---|---|---|---|---|
| | Proportion | Std. Error | 95% confidence interval | |
| | | | Lower | Upper |
| OBC | 0.459 | 0.001 | 0.457 | 0.461 |
| Others | 0.204 | 0.001 | 0.202 | 0.206 |
| **Wealth Quintile** | | | | |
| Poorest | 0.251 | 0.001 | 0.249 | 0.252 |
| Poorer | 0.218 | 0.001 | 0.216 | 0.219 |
| Middle | 0.197 | 0.001 | 0.196 | 0.199 |
| Richer | 0.183 | 0.001 | 0.182 | 0.185 |
| Richest | 0.151 | 0.001 | 0.150 | 0.153 |
| **Child's vaccination** | | | | |
| No | 0.086 | 0.001 | 0.085 | 0.087 |
| Partial | 0.401 | 0.001 | 0.399 | 0.403 |
| Full | 0.513 | 0.001 | 0.511 | 0.515 |

associated with birth registration. The model 3 result showed random variance of 1.21, 0.26 and 3.29 at the state, district and individual levels respectively. Moreover, ICC value of 0.25 at the state level showed that 25 percent of total variation in birth registration level is explained by between state level differences while the remaining 75 percent lies within states. Besides, ICC value of 0.06 at the district level indicated that 6 percent of total variation in birth registration level lie between districts, indicates a need for adequate programmes to increase birth registration level at lower administrative level.

We found as compared with children age 0–1 year, children between age group 1–3 years and 3–5 years have higher (AOR = 1.14, 95% CI [1.11, 1.18], p<0.001) and lower (AOR = 0.96, 95% CI [0.93, 0.99], p = 0.04) likelihood of birth registration. Interestingly, female children have a higher likelihood of birth registration as compared to male children (AOR = 1.04, 95% CI [1.02, 1.06], p<0.001). Besides, children of birth order 2 (AOR = 0.78, 95% CI [0.76, 0.80], p<0.001] and 3 (AOR = 0.68, 95% CI [0.65, 0.70], p<0.001) showed lower likelihood of birth registration as compared with children of birth order 1. Place of resident was found to be significantly associated with child's birth registration. We found as compared with children living in urban areas, children living in rural areas have a lower likelihood of birth registration (AOR = 0.85, 95% CI [0.82, 0.88], p<0.001).

Mother's characteristics were found to be significantly associated with child's birth registration. As compared with mothers among younger age group (15–24) years, mothers among age group 25–34 years (AOR = 1.13, 95% CI [1.10, 1.66], p<0.001) and 35–49 years (AOR = 1.21, 95% CI [1.15, 1.26], p<0.001) have a higher likelihood of birth registration for their children. A lower odds of birth registration was observed among primary educated (AOR = 0.74, 95% CI [0.69, 0.79], p<0.001) and illiterate mothers (AOR = 0.57, 95% CI [0.54, 0.61], P<0.001) as compared to higher educated mothers. Besides, mothers who had exposed to partial (AOR = 0.91, 95% CI [0.86, 0.96], p<0.001) and no (AOR = 0.78, 95% CI [0.74, 0.83], P<0.001) media showed a lower likelihood for their child's birth registration as compared with mothers exposed to full media. Wealth status of household was also found to be significantly associated. We found as compared with mothers among richest wealth quintile, mothers belong to poorest (AOR = 0.38, 95% CI [0.36, 0.41], p<0.001) and poorer (AOR = 0.48, 95% CI [0.46, 0.51], p<0.001) wealth quintile showed a lower likelihood for birth registration of their children.

**Table 2. The percent of children under age five years whose birth has ever been registered by baseline characteristics, NFHS-2015-16, India.**

| Independent variables | Percent | Frequency | $\chi^2$ value | P-value |
|---|---|---|---|---|
| **Sex of child** | | | | |
| Male | 79.91 | 114095 | 7.4 | 0.025 |
| Female | 80.53 | 104540 | | |
| **Child's age (in year)** | | | | |
| 0–1 | 79.56 | 41845 | | |
| 1–3 | 81.75 | 87487 | 202.3 | <0.001 |
| 3–5 | 79.00 | 89303 | | |
| **Birth Order** | | | | |
| 1 | 86.91 | 82070 | | |
| 2 | 82.97 | 70917 | 7800.0 | <0.001 |
| 3 | 74.44 | 33995 | | |
| 3+ | 62.83 | 31653 | | |
| **Place of residence** | | | | |
| Urban | 89.14 | 62684 | 3300.0 | <0.001 |
| Rural | 76.62 | 155951 | | |
| **Mother's age** | | | | |
| 15–24 | 79.56 | 41815 | | |
| 25–34 | 81.75 | 87487 | 202.3 | <0.001 |
| 35–49 | 79.00 | 89333 | | |
| **Mother's education** | | | | |
| Illiterate | 64.35 | 65897 | | |
| Primary | 79.69 | 30420 | 14000.0 | <0.001 |
| Secondary | 88.18 | 99535 | | |
| Higher | 91.95 | 22783 | | |
| **Marital Status** | | | | |
| Currently married | 80.24 | 216153 | 0.3 | 0.855 |
| Separated/divorced/widow | 77.39 | 2482 | | |
| **Media exposure** | | | | |
| No | 64.62 | 58833 | | |
| Partial | 85.49 | 144386 | 10000.0 | 0.001 |
| All | 90.19 | 15416 | | |
| **Sex of the head of household** | | | | |
| Male | 80.62 | 192622 | | |
| Female | 77.20 | 26013 | 25.0 | 0.001 |
| **Religion** | | | | |
| Hindus | 80.15 | 171570 | | |
| Muslims | 77.87 | 36312 | 257.4 | <0.001 |
| Others | 88.98 | 10753 | | |
| **Caste** | | | | |
| SC | 79.28 | 47138 | | |
| ST | 76.00 | 23252 | 1200.0 | <0.001 |
| OBC | 78.09 | 95844 | | |
| others | 86.45 | 52401 | | |
| **Wealth Quintile** | | | | |
| Poorest | 64.32 | 54797 | | |
| Poorer | 77.92 | 47606 | | |
| Middle | 84.65 | 43144 | 14000.0 | <0.001 |

*(Continued)*

**Table 2.** (Continued)

| Independent variables | Percent | Frequency | $\chi^2$ value | P-value |
|---|---|---|---|---|
| Richer | 88.97 | 40044 | | |
| Richest | 94.44 | 33044 | | |
| **Child's vaccination** | | | | |
| No | 61.70 | 18885 | | |
| Partial | 77.17 | 87609 | 6900.0 | <0.001 |
| Full | 85.70 | 112141 | | |
| **Total** | **80.21** | **218635** | | |

Note:- Total refers total weighted frequency.

The result also showed that living in a district with higher proportion of SCs households increases the odds of child's birth registration (AOR = 1.00, 95% CI [0.99, 1.01], p = 0.12). However, we found the result was not significant. Moreover, residing in a district with higher proportion of institutional birth (AOR = 1.01, 95% CI [1.00, 1.01], p<0.001) and proportion of vaccinated children (AOR = 1.01, 95% CI [1.00, 1.01], p<0.001) showed a higher odds of child's birth registration.

## Discussion

Birth registration is a legal process, but it is essential for proving fundamental rights and essential services such as education and health facility to children. It protects children from unlawful activities such as child labour, trafficking, and child marriage. Registration of Birth and Death (RBD) act, 1969, mandates registration of all births and deaths within 21 days of the event [13]. There has been significant improvement in coverage of birth registration in the last ten years. The registration level increased to 80% in 2016 from 41% in 2005 [14, 19]. However, there is an uneven improvement in birth registration across the nation and within states. The existing studies were based on small sample size, primarily focussed on administrative challenges, system's design, and need and benefits associated with functional system [18]. Moreover, previous studies documented predictors of birth registration mostly in the context of other nations [10, 16, 17]. However, this study presents a multilevel analysis at state, district and individual level and spatial mapping of birth registration level in India. This study shows 25 percent of total variation in child's birth registration level in India lies between states and remaining 75 percent variation lie within states, indicates need for adequate policies for improving birth registration level at lower administrative level. We found out of 640 districts, 254 districts in India where the registration level was below the national average (80.21%). This study found demographic, socioeconomic, and healthcare variables are significant covariates of child's birth registration.

This study showed higher birth registration among female children. A higher birth registration among female children could be attributed to financial benefits schemes such as Balika Samridhi Yojna. Each female child is entitled to 500 rupees post-birth and receives a scholarship from India's Government to complete a set of years of schooling [20]. A previous study also showed a positive impact of cash transfer on children's birth registration. The cash transfer scheme increased the percent of registered female children to 39 percent from 24 percent in Assam [21]. Children among age groups 1–3 are more likely to register than children who have not completed their first birthday. Parents don't register their children until they seek school admission of their child. Previous studies also documented likelihood of child's birth registration level increases with increase in child's age [10, 15, 17].

**Table 3. Results of multilevel binary logistic regression of demographic, socioeconomic, and healthcare factors associated with birth registration, India.**

| | Null Model | | | Model 1 | | | Model 2 | | | Model 3 | | |
|---|---|---|---|---|---|---|---|---|---|---|---|---|
| | | 95% C.I | | | 95% C.I | | | 95% C.I | | | 95% C.I | |
| Fixed effect parameter | OR | Lower | Upper | AOR | Lower | Upper | AOR | Lower | Upper | AOR | Lower | Upper |
| Intercept | 14.73*** | 9.85 | 22.01 | 39.17*** | 25.35 | 60.52 | 50.30*** | 32.93 | 76.81 | 10.03*** | 6.04 | 16.67 |
| **Child's age (Years)** | | | | | | | | | | | | |
| 0–1 | Reference | | | | | | | | | | | |
| 1–3 | | | | 1.11*** | 1.07 | 1.16 | 1.14*** | 1.11 | 1.18 | 1.14*** | 1.11 | 1.18 |
| 3–5 | | | | 0.91*** | 0.87 | 0.95 | 0.96** | 0.93 | 0.99 | 0.96* | 0.93 | 0.99 |
| **Sex of the child** | | | | | | | | | | | | |
| Male | Reference | | | | | | | | | | | |
| Female | | | | 1.03** | 1.01 | 1.05 | 1.04*** | 1.02 | 1.06 | 1.04*** | 1.02 | 1.06 |
| **Birth order** | | | | | | | | | | | | |
| 1 | Reference | | | | | | | | | | | |
| 2 | | | | 0.72*** | 0.69 | 0.75 | 0.78*** | 0.76 | 0.80 | 0.78*** | 0.76 | 0.80 |
| 3 | | | | 0.56*** | 0.53 | 0.58 | 0.68*** | 0.65 | 0.70 | 0.68*** | 0.65 | 0.70 |
| 3+ | | | | 0.41*** | 0.39 | 0.43 | 0.58*** | 0.56 | 0.60 | 0.58*** | 0.56 | 0.61 |
| **Place of residence** | | | | | | | | | | | | |
| Urban | Reference | | | | | | | | | | | |
| Rural | | | | 0.61*** | 0.58 | 0.63 | 0.85*** | 0.82 | 0.88 | 0.85*** | 0.82 | 0.88 |
| **Mother's age (years)** | | | | | | | | | | | | |
| 15–24 | Reference | | | | | | | | | | | |
| 25–34 | | | | 1.18*** | 1.14 | 1.23 | 1.13*** | 1.10 | 1.16 | 1.13*** | 1.10 | 1.16 |
| 35–49 | | | | 1.23*** | 1.18 | 1.28 | 1.21*** | 1.15 | 1.27 | 1.21*** | 1.15 | 1.26 |
| **Mother's education** | | | | | | | | | | | | |
| Higher | Reference | | | | | | | | | | | |
| Middle | | | | | | | 0.86*** | 0.82 | 0.92 | 0.86*** | 0.81 | 0.91 |
| Primary | | | | | | | 0.88*** | 0.85 | 0.90 | 0.74*** | 0.69 | 0.79 |
| Illiterate | | | | | | | 0.74*** | 0.69 | 0.78 | 0.57*** | 0.54 | 0.61 |
| **Media Exposure** | | | | | | | | | | | | |
| Full | Reference | | | | | | | | | | | |
| Partial | | | | | | | 0.91*** | 0.86 | 0.96 | 0.91*** | 0.86 | 0.96 |
| No | | | | | | | 0.78*** | 0.73 | 0.83 | 0.78*** | 0.74 | 0.83 |
| **Wealth quintile** | | | | | | | | | | | | |
| Richest | Reference | | | | | | | | | | | |
| Richer | | | | | | | 0.69*** | 0.65 | 0.73 | 0.69*** | 0.65 | 0.73 |
| Middle | | | | | | | 0.59*** | 0.56 | 0.63 | 0.59*** | 0.56 | 0.63 |
| Poorer | | | | | | | 0.48*** | 0.45 | 0.51 | 0.48*** | 0.46 | 0.51 |
| Poorest | | | | | | | 0.38*** | 0.36 | 0.41 | 0.38*** | 0.36 | 0.41 |
| **Religion** | | | | | | | | | | | | |
| Hindus | Reference | | | | | | | | | | | |
| Muslims | | | | | | | 0.90*** | 0.87 | 0.93 | 0.90*** | 0.87 | 0.94 |
| Others | | | | | | | 0.79*** | 0.74 | 0.85 | 0.80*** | 0.75 | 0.86 |
| **District level variable** | | | | | | | | | | | | |
| Proportion SCs | | | | | | | | | | 1.006 | 0.99 | 1.01 |
| Proportional Institutional Birth | | | | | | | | | | 1.01*** | 1.00 | 1.01 |
| Proportion Children vaccinated | | | | | | | | | | 1.01*** | 1.00 | 1.01 |
| **Random effect parameter** | | | | | | | | | | | | |
| District | 0.40 | 0.63 | | 0.36 | 0.60 | | 0.32 | 0.56 | | 0.26 | 0.51 | |

*(Continued)*

**Table 3.** (Continued)

| Fixed effect parameter | Null Model | | | Model 1 | | | Model 2 | | | Model 3 | | |
|---|---|---|---|---|---|---|---|---|---|---|---|---|
| | | 95% C.I | | | 95% C.I | | | 95% C.I | | | 95% C.I | |
| | OR | Lower | Upper | AOR | Lower | Upper | AOR | Lower | Upper | AOR | Lower | Upper |
| State | 2.05 | 1.43 | | 1.81 | 1.34 | | 1.55 | 1.24 | | 1.21 | 1.10 | |
| Residual | 3.29 | 1.81 | | 3.29 | 1.81 | | 3.29 | 1.81 | | 3.29 | 1.81 | |
| **ICC** | | | | | | | | | | | | |
| District | 0.07 | 0.26 | | 0.07 | 0.26 | | 0.06 | 0.24 | | 0.06 | 0.24 | |
| State | 0.36 | 0.60 | | 0.33 | 0.57 | | 0.30 | 0.54 | | 0.25 | 0.50 | |
| AIC | 191395 | | | 187588 | | | 183570 | | | 183490 | | |
| Log-likelihood | -95694 | | | -93782 | | | -91762 | | | -91719 | | |
| No of group in districts | 640 | | | 640 | | | 640 | | | 640 | | |
| Number of groups in State | 36 | | | 36 | | | 36 | | | 36 | | |
| Observations | 225867 | | | 225867 | | | 225867 | | | 225867 | | |
| VIF (mean) | | | | 1.33 | | | 1.37 | | | 1.34 | | |

Note:-

*** $p < 0.001$;

** $p < 0.01$;

* $p < 0.05$

Children in rural areas are less likely to register, which is not unusual; the distance to the registration center includes higher financial and indirect opportunity costs for the family. A higher proportion of institutional birth results to higher registered birth in urban areas. The higher birth registration among institutional births could be attributed to the fact that it is the duty of medical officers to register all births delivered in the health facilities. Previous studies also supported our finding [3, 10, 13, 22]. Living in a district with higher proportion of institutional birth increases the odds of child's birth registration. Institutional birth would increase a mother's awareness regarding child care, including timely immunisation and birth registration. A study showed a higher probability of birth registration for institutionally delivered children in Latin America and Carribean [23].

This study showed that younger mothers (15–24 years) have a lower likelihood of birth registration of their children. A lower practice of birth registration among young mothers could be attributed to less awareness of the registration process and a childcare experience. This study found a significant positive association between children's birth registration and the mother's education level. Mothers who completed formal years of education have more access to institutional health care, media exposure, and knowledge on the registration process. Besides, an educated woman has exposure to social network of the other educated person which increases the odds of her child's birth registration. Previous studies documented a similar finding [3, 8, 10, 16, 17, 24]. A higher proportion of children are registered among currently married mothers, however, result was not significant in our study. Previous studies showed caring for children by both parents may affect the quality of care and children's well-being [16, 17, 25]. Also, Unicef documented that in many countries, a single mother cannot register her child [3]. In India's strong patriarchal societies, a woman finds difficult to register her child's without mentioning father's name. However, the birth registration law is not mandatory require father's name for a child's birth registration. A study showed that counties where the law gives equal right to women to report birth of their child to civil authority, however, patriarchal attitudes and discriminatory practices against women affect their ability to do so [26].

Exposure to different forms of media is significant for gaining knowledge and awareness regarding various government laws and schemes. This study also shows a strong association of media exposure among mothers and their children's birth registration, consistent with other studies [8, 13]. As evident in previous literature, children who belong to underprivileged groups such as Muslims and STs have lower birth registration [10, 16, 24]. A study showed lower utilisation of safe delivery care among muslims women which could contribute to a lower odds of their child birth registration [15]. In particular culture or community, more preference is given to traditional norms (such as name ceremony) rather than formal birth registration of children. Also, minority groups like some tribal people in India are more likely to live in remote areas where access to birth registration services is complicated. Further, children among richer and richest households have a higher likelihood of birth registration [3, 8, 22]. This finding is not an exception with our study. The direct and indirect cost associated with registration is a barrier to birth registration [16, 27]. Loss of wage due to away from the work and transportation cost hindered poorer households to registering their children's birth to civil authority. In addition, late birth registration fees and affidavit from a notary public is required if birth are registered after 21 days which could be discouraging factors for poorer households. There is a need to open register centre close to communities and use of mobile services for registration in far flung areas, may reduce tavel cost and motivate people to register their children within the prescribed time limit.

Moreover, this study showed significant positive association between the child's vaccination and his/her birth registration. Living in a district with higher proportion of immunised children significantly increases the odds of birth registration. Considering the fact, registration centers are open within some health care facilities may have contributed to this finding. A previous study documented vaccination services provide an opportunity for health workers to be alerted to absence of birth certificate, leading vaccination to be viewed as potential driver to register a child's birth. Besides, another study showed that in 43 countries having a vaccination card makes children more likely to be registered [10]. This study's findings provide a way forward towards improving the level of birth registration and focused intervention on overcoming existing barriers.

Despite a comprehensive analysis, this study has some limitations. The quality of findings may be affected by recall or reporting biases. Mainly, possession of the birth certificate is socially desirable behavior and may lead to overestimating registered birth; however, birth certificate was requested to confirm the reporting. Also, this study examined the association between birth registration and the independent variables as it is based on cross sectional data, it does not establish causal link.

## Conclusions

Birth registration provides access to government-aided essential services such as healthcare, education, and legal protection. Health officials and policymakers frequently use birth registration data in framing health policies and socioeconomic development programs. Birth registration level vary significantly between and within states, indicates need for the adequate programmes at lower administrative level. Children's age, birth order, place of residence, religious affiliation, and vaccination appear to be significant determinants of birth registration. We strongly suggest linking the birth registration facilities with health institutions. We also suggest periodic awareness campaigns on birth registration benefits among underprivileged population groups and low-performing districts. Establishing a community-based birth registration unit, i.e., registration unit at primary and community healthcare, may ensure accessibility and improve birth registration completeness.

## Supporting information

**S1 Table. Observed birth registration and predicted estimates of birth registration level (%) by districts of India, 2015–16.**
(XLSX)

## Acknowledgments

The authors would like to acknowledge all the Ambedkar Library staff, Jawaharlal Nehru University, New Delhi, for providing access to Journals.

## Author Contributions

**Conceptualization:** Nandita Saikia.

**Formal analysis:** Krishna Kumar.

**Supervision:** Nandita Saikia.

**Writing – original draft:** Krishna Kumar.

**Writing – review & editing:** Nandita Saikia.

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
