## [Decision Letter · Decision Letter 0]

31 May 2021

PONE-D-21-06116

Determinants of Birth Registration in India: Evidence from NFHS 2015-16

PLOS ONE

Dear Dr. Kumar,

Thank you for submitting your manuscript to PLOS ONE. After careful consideration, we feel that it has merit but does not fully meet PLOS ONE’s publication criteria as it currently stands. Therefore, we invite you to submit a revised version of the manuscript that addresses the points raised during the review process.

We look forward to receiving your revised manuscript.

Kind regards,

Kannan Navaneetham, PhD

Academic Editor

PLOS ONE

Journal Requirements:

2. Please refer to the specific statistical analyses performed as well as any post-hoc corrections to correct for multiple comparisons. If these were not performed please justify the reasons. Please refer to our statistical reporting guidelines for assistance (https://journals.plos.org/plosone/s/submission-guidelines.#loc-statistical-reporting). Additionally, please ensure you have thoroughly discussed any potential limitations of this study within the Discussion.

3. Please correct your reference to "p=0.000" to "p<0.001" or as similarly appropriate, as p values cannot equal zero.

Reviewers' comments:

Reviewer's Responses to Questions

**Comments to the Author**

1. Is the manuscript technically sound, and do the data support the conclusions?

Reviewer #1: Yes

Reviewer #2: Partly

2. Has the statistical analysis been performed appropriately and rigorously? 

Reviewer #1: Yes

Reviewer #2: No

3. Have the authors made all data underlying the findings in their manuscript fully available?

Reviewer #1: Yes

Reviewer #2: Yes

4. Is the manuscript presented in an intelligible fashion and written in standard English?

Reviewer #1: Yes

Reviewer #2: No

5. Review Comments to the Author

Reviewer #1: The manuscript was well written and the statistical analysis is sound. Below is the minor correction that the authors should make.

1. P values of 0.000 should be written as p < 0.001 just like appeared in notes of the results

2. “insignificant” should read “not significant” (lines 276 and 280)

Reviewer #2: I understand that this paper studies the determinants of birth registration in India using a nationally representative and the latest round of NFHS 2015-16. The authors carried out the bivariate analysis and multilevel binary logistic regression to identify significant covariates at the individual, district, and state levels to determine the likelihood of birth registration. Authors also did spatial mapping to present the status of birth registration across districts in India using GIS.

However, I have concerns that the manuscript is not written well to merit publication in PlosOne. Additionally, I have concerns over the methodology and presentation of this manuscript. I have listed my concerns below:

1. To start with, I believe the abstract is not written well. It lacks the standard presentation style and is not very smooth to read. I believe in the very first statement the authors need to present the research problem clearly and succinctly. And, in the next sentence the authors need to write why it is important to study the problem. Then the data description, methodology, results, and conclusion need to follow in a nice and smooth manner. I feel while everything is there the writing lacks a coherent presentation style. More importantly the authors need to bring in what is the significant value addition of this research, which I feel is overall lacking.

2. Overall, the manuscript is not well written. The background sections are not well motivated and lacks a robust literature review. Introduction seems very disjoint. It fails to captivate the reader into the topic. I recommend rewriting the background section with a strong literature review that will motivate the study with a nice flow in writing. Also, the authors fail to review one important previous study on this topic on India that was published on PLoS ONE, Mohanty and Gebremedhin (2018) that I believe is an important precursor for this manuscript. Please see below:

Mohanty I, Gebremedhin TA (2018) Maternal autonomy and birth registration in India: Who gets counted? PLoS ONE 13(3): e0194095.

https://doi.org/10.1371/journal.pone.0194095

3. In the final paragraph of the Introduction section (line no:108-109) the authors identified the gap that few previous studies attempted to investigate the civil registration systems design and functional status. However, all through the manuscript there was no further discussion on this issue nor did the authors made any attempt to include any variables in the model that would have represented differences in the Civil registration systems design and function/practice across districts/states in India. Also, on several other places in the manuscript, I find similar orphan sentences initiating a discussion or reporting a result that were not closed properly.

4. On line no: 109-111, the authors wrote, there is a lack of systematic research examining predictors of birth registration in India at an individual and community level while they have failed to refer the publication Mohanty and Gebremedhin (2018), which is an important pre-cursor of this study and the authors need to bring out comparison between the present study and Mohanty Gebremedhin (2018) highlighting the significant added value of this manuscript.

5. On line no: 119 on Materials and methods section (Data Source) the authors need to rephrase writing – they have used the most recent round of a nationally representative demographic and Health Survey on India, National Family Health Survey, 2015-16 (NFHS-4).

6. In the Study design and samples section, I disagree with the authors decision to select the sample only for the districts (n=258) where the birth registration level was lower than the national average. This action may have led to significant bias in the regression results while the model would fail to identify the facilitators (motivating factors) that positively influence the birth registration. I believe this is an important lacking in the model that the authors need to address to qualify their manuscript for publication. This is a significant methodological failing if there are no other systemic or contextual differences between the districts where the birth registration level was lower than the national average and where it was higher. If the authors are interested to study the difference between these two groups of districts, they can choose to include an indicator variable in the model and interact that indicator variable with other important variables to study the differential effect. However, presently in this shape the model suffers from sample selection bias.

7. Also, I believe it would be useful if the authors present what proportion of the dependent variable represent children who have a birth certificate compared to those who are ever been registered by the civil authority but, do not have a birth certificate.

8. I believe it is useful that the authors generated a map of level of birth registration of children under age five years, Indian districts, 2016 using GIS. However, it would be useful and will add significant value to the research undertaken in this manuscript if the authors generate another map with the multi-level logistic regression model’s predicted estimates and compare the two maps.

9. The Table 1 where the authors presented their bivariate analysis, is redundant and does not improve the overall presentation of the manuscript since the authors did run a multi-level logistic regression model in the later part and presented the results in Table 2. However, it is required that the authors present a descriptive statistic table (including the means/proportions, standard deviation, minimum and maximum values of the dependent and independent variables with the full sample size) of the variables included in the final model. This is useful information for the readers to assess the model.

10. On line no: 225-227, the authors reported that about 59% of children are registered among no vaccinated children, whereas 71% of children are registered among children who received at least one vaccine. And the vaccination status is later coming up as statistically significant in the bivariate and multi-variate regression models. However, I believe it raises an important question here, which one comes first in the sequence of occurrence, or which one is the cause? Do the families need to register the birth first and then go for vaccination or, it is other way round? It is important to bring in what is required and what is practice and if there is a variation in practice across districts/states. Also, is there an endogeneity issue here?

11. On line no: 229-231 the authors reported that nearly 64% of children are registered among currently married mothers, whereas only 58% of children were registered among divorced or widowed marital status. I believe these figures are misleading without an idea on the descriptive statistics of the sample. For example, it will be useful to interpret these statistics with reference to what proportion of the children in the sample belongs to a single mother with a divorce/widow status. The same argument applies to all the descriptions on Table 1, which I believe is redundant.

12. In Table 2 in presenting the results of the multi-level logistic regression analysis the authors presented the coefficients and not the odds ratios. I believe presenting odds ratios are the accepted standard for logistic regression.

13. The manuscript is lacking a discussion on the list of variables that were included in the multi-level regression analysis at different levels and the proportion of variations in the model explained by variables included at different levels. For example, if within district differences account for most of the variation in the model or is it within states/individuals. A discussion on ICC or Variance Partition Coefficient (VPC) to represent the percentage variance explained by the different levels and it’s policy implications would be useful.

14. Again, the discussion section is poorly written, and it is not clearly bringing out the significant added value of this study over previous studies. In most places the authors present their results and then in the next sentence they are saying previous literature showed similar findings. I believe the authors need to discuss their results with supporting literature and possible explanations and greater policy implications.

15. On line no: 320 it is reported that in many countries, a single mother cannot register their children. Since this manuscript is on India, I believe the authors need to bring out a discussion on India.

16. On line no: 320-321 the authors need to explain what do they mean by different media kinds and how do they construct/define this variable in their regression.

17. On line no: 329, the authors wrote that the cost associated with registration is a barrier to birth registration. I believe they need to discuss this further, which cost – direct/indirect.

18. Overall the manuscript needs to be grossly rewritten with a strong literature review and the major issues in the methodology, presentation and mapping and discussion sections need to

6. PLOS authors have the option to publish the peer review history of their article (what does this mean?). If published, this will include your full peer review and any attached files.

Reviewer #1: **Yes: **Alphonsus Isara

Reviewer #2: No

---

## [Author Response · Author response to Decision Letter 0]

2 Aug 2021

Response to reviewers

Reviewer #1:

The manuscript was well written and the statistical analysis is sound. Below is the minor correction that the authors should make.

1. P values of 0.000 should be written as p < 0.001 just like appeared in notes of the results.

Response- We thank you for this suggestion. Now, p-value of 0.000 has been changed to p<0.001

2. “insignificant” should read “not significant” (lines 276 and 280)

Response- We thank you for this suggestion. We changed insignificant to not significant.

Reviewer #2: 

I understand that this paper studies the determinants of birth registration in India using a nationally representative and the latest round of NFHS 2015-16. The authors carried out the bivariate analysis and multilevel binary logistic regression to identify significant covariates at the individual, district, and state levels to determine the likelihood of birth registration. Authors also did spatial mapping to present the status of birth registration across districts in India using GIS.

However, I have concerns that the manuscript is not written well to merit publication in PlosOne. Additionally, I have concerns over the methodology and presentation of this manuscript. I have listed my concerns below:

1. To start with, I believe the abstract is not written well. It lacks the standard presentation style and is not very smooth to read. I believe in the very first statement the authors need to present the research problem clearly and succinctly. And, in the next sentence the authors need to write why it is important to study the problem. Then the data description, methodology, results, and conclusion need to follow in a nice and smooth manner. I feel while everything is there the writing lacks a coherent presentation style. More importantly the authors need to bring in what is the significant value addition of this research, which I feel is overall lacking.

Response: We have revised the abstract as per your suggestions.

2. Overall, the manuscript is not well written. The background sections are not well motivated and lacks a robust literature review. Introduction seems very disjoint. It fails to captivate the reader into the topic. I recommend rewriting the background section with a strong literature review that will motivate the study with a nice flow in writing. Also, the authors fail to review one important previous study on this topic on India that was published on PLoS ONE, Mohanty and Gebremedhin (2018) that I believe is an important precursor for this manuscript. Please see below:

Mohanty I, Gebremedhin TA (2018) Maternal autonomy and birth registration in India: Who gets counted? PLoS ONE 13(3): e0194095.

https://doi.org/10.1371/journal.pone.0194095

Response: Thank you for your observation and wonderful suggestion. We revised the background section and added a few important literature to make it more informative. We referred the study you mentioned. 

3. In the final paragraph of the introduction section (line no:108-109), the authors identified the gap that few previous studies attempted to investigate the civil registration systems design and functional status. However, all through the manuscript there was no further discussion on this issue nor did the authors made any attempt to include any variables in the model that would have represented differences in the Civil registration systems design and function/practice across districts/states in India. Also, on several other places in the manuscript, I find similar orphan sentences initiating a discussion or reporting a result that were not closed properly.

Response: I thank you for this observation. NFHS does not provide information on quality and function status of civil registration system. Therefore, we could not include such variables in the model. Besides, this study primarily focuses on covariates of birth registration. Furthermore, I have revised the discussion section and supported our results with previous literature.

4. On line no: 109-111, the authors wrote, there is a lack of systematic research examining predictors of birth registration in India at an individual and community level while they have failed to refer the publication Mohanty and Gebremedhin (2018), which is an important pre-cursor of this study and the authors need to bring out comparison between the present study and Mohanty Gebremedhin (2018) highlighting the significant added value of this manuscript.

Response: I thank you for this important observation. We have reviewed the suggested literature (Mohanty and Gebremedhin, 2018). The literature included demographic, socioeconomic and women empowerment variables, examined the variation in birth registration level at the district and community level. However, the literature was primarily focussed on the variables representing mother’s social and economic, and bargaining power in the household and its association with her child’s birth registration. The study did not present birth registration percent at district level due to small sample size. In addition, the referred study was based on 2011-12 and may not present true situation regarding birth registration level. On the other hand, our study included a large sample size (225867) from a recent large-scale survey, NFHS-2015-16, may provide robust results useful for decision making to increase birth registration level in India and its states and districts. In addition to multi-level modelling, our study showed spatial mapping of birth registration level by district of India. Our study showed considerable variation in birth registration level due to the state and district level factors, demographic, socioeconomic, and health care characteristics that is still not widely explored in India.

5. On line no: 119 on Materials and methods section (Data Source) the authors need to rephrase writing – they have used the most recent round of a nationally representative demographic and Health Survey on India, National Family Health Survey, 2015-16 (NFHS-4).

Response: Thank you for this suggestion. We have rephrased the sentence now. (please see line no. 129)

6. In the Study design and samples section, I disagree with the authors decision to select the sample only for the districts (n=258) where the birth registration level was lower than the national average. This action may have led to significant bias in the regression results while the model would fail to identify the facilitators (motivating factors) that positively influence the birth registration. I believe this is an important lacking in the model that the authors need to address to qualify their manuscript for publication. This is a significant methodological failing if there are no other systemic or contextual differences between the districts where the birth registration level was lower than the national average and where it was higher. If the authors are interested to study the difference between these two groups of districts, they can choose to include an indicator variable in the model and interact that indicator variable with other important variables to study the differential effect. However, presently in this shape the model suffers from sample selection bias.

Response: We thank you for this suggestion. We have revised the analysis for all districts of India now. (Total sample size=225867, district=640, state=36). 

7. Also, I believe it would be useful if the authors present what proportion of the dependent variable represent children who have a birth certificate compared to those who are ever been registered by the civil authority but, do not have a birth certificate.

Response: Thank you for this suggestion. We have added a descriptive statistics table for a dependent and all independent variables (Table 1). Table 1 shows proportion of children who have birth certificate and ever registered children but do not have birth certificate.

8. I believe it is useful that the authors generated a map of level of birth registration of children under age five years, Indian districts, 2016 using GIS. However, it would be useful and will add significant value to the research undertaken in this manuscript if the authors generate another map with the multi-level logistic regression model’s predicted estimates and compare the two maps.

Response: Thank you for your suggestion. We have incorporated this important suggestion. We have added another map of multi-level logistic regression model’s predicted estimates of birth registration level. Also, we have compared the two maps in the revised manuscript (Figs 1 and 2).

9. The Table 1 where the authors presented their bivariate analysis is redundant and does not improve the overall presentation of the manuscript since the authors did run a multi-level logistic regression model in the later part and presented the results in Table 2. However, it is required that the authors present a descriptive statistic table (including the means/proportions, standard deviation, minimum and maximum values of the dependent and independent variables with the full sample size) of the variables included in the final model. This is useful information for the readers to assess the model..

Response: Thank you for this important suggestion. Now, we have included descriptive statistics of dependent and independent variables with full sample size. However, we showed proportion, standard errors and confidence interval for each variable because the variables included in our study are categorical in nature.

10. On line no: 225-227, the authors reported that about 59% of children are registered among no vaccinated children, whereas 71% of children are registered among children who received at least one vaccine. And the vaccination status is later coming up as statistically significant in the bivariate and multi-variate regression models. However, I believe it raises an important question here, which one comes first in the sequence of occurrence, or which one is the cause? Do the families need to register the birth first and then go for vaccination or, it is other way round? It is important to bring in what is required and what is practice and if there is a variation in practice across districts/states. Also, is there an endogeneity issue here?

Response: Usually, birth registration is done at the time of birth and vaccinations are done afterwards according to vaccination schedules. Because of our study is of cross -sectional in nature, information on both vaccination and registration status was collected at the time of survey. Our study shows partial and fully vaccinated children are more likely to get registered their birth, but because of lack of timing of birth registration data, we cannot infer which one causes which. We can simply infer the association between vaccination and birth registration. Previous studies also showed vaccination drive alert health care worker to the absence of a birth certificate (UNICEF, 2005). India’s health policy does not mandatorily require birth certificate of children to vaccinate them; therefore, birth certificate should not directly affect child’s vaccination. We have mentioned this point in the limitation of the study (line no. 434-436). 

11. On line no: 229-231 the authors reported that nearly 64% of children are registered among currently married mothers, whereas only 58% of children were registered among divorced or widowed marital status. I believe these figures are misleading without an idea on the descriptive statistics of the sample. For example, it will be useful to interpret these statistics with reference to what proportion of the children in the sample belongs to a single mother with a divorce/widow status. The same argument applies to all the descriptions on Table 1, which I believe is redundant.

Response: We thank you for this observation and suggestion. Now, we have included descriptive statistics of a dependent and independent variables in the revised manuscript (Table 1).

12. In Table 2 in presenting the results of the multi-level logistic regression analysis the authors presented the coefficients and not the odds ratios. I believe presenting odds ratios are the accepted standard for logistic regression.

Response: We revised the model now, and have presented odds ratio, confidence interval and p-value for each explanatory variable included in the model (Please see table 3).

13. The manuscript is lacking a discussion on the list of variables that were included in the multi-level regression analysis at different levels and the proportion of variations in the model explained by variables included at different levels. For example, if within district differences account for most of the variation in the model or is it within states/individuals. A discussion on ICC or Variance Partition Coefficient (VPC) to represent the percentage variance explained by the different levels and it’s policy implications would be useful.

Response: We included discussion on ICC and other district level explanatory variables in the revised manuscript. (Please see line no. 353-356, 377-380, 421-426).

14. Again, the discussion section is poorly written, and it is not clearly bringing out the significant added value of this study over previous studies. In most places, the authors present their results and then in the next sentence they are saying previous literature showed similar findings. I believe the authors need to discuss their results with supporting literature and possible explanations and greater policy implications.

Response: We have revised the discussion section in the light of your comments.

15. On line no: 320 it is reported that in many countries, a single mother cannot register their children. Since this manuscript is on India, I believe the authors need to bring out a discussion on India.

Response: We have added a few points on marital status and birth registration in the context of India (line no. 391-399). However, we have not found any published study based on India’s dataset that clearly highlights association between marital status and child’s birth registration.

16. On line no: 320-321 the authors need to explain what do they mean by different media kinds and how do they construct/define this variable in their regression.

Response: We have defined the different forms of media, how we constructed this variable in the material and methods section of the revised manuscript (line no. 165-171).

17. On line no: 329, the authors wrote that the cost associated with registration is a barrier to birth registration. I believe they need to discuss this further, which cost – direct/indirect.

Response: We have included that both direct (late fine, affidavit cost) and indirect cost (loss of one day wage, travelling cost) is barriers to birth registration (line no. 411-415).

18. Overall the manuscript needs to be grossly rewritten with a strong literature review and the major issues in the methodology, presentation and mapping and discussion sections need to

Response: We have revised the manuscript on the basis of your suggestions. We sincerely hope that it will be considered for publication now.

---

## [Decision Letter · Decision Letter 1]

23 Aug 2021

Determinants of Birth Registration in India: Evidence from NFHS 2015-16

PONE-D-21-06116R1

Dear Dr. Kumar,

We’re pleased to inform you that your manuscript has been judged scientifically suitable for publication and will be formally accepted for publication once it meets all outstanding technical requirements.

Kind regards,

Kannan Navaneetham, PhD

Academic Editor

PLOS ONE

Additional Editor Comments (optional):

Reviewers' comments:

Reviewer's Responses to Questions

**Comments to the Author**

1. If the authors have adequately addressed your comments raised in a previous round of review and you feel that this manuscript is now acceptable for publication, you may indicate that here to bypass the “Comments to the Author” section, enter your conflict of interest statement in the “Confidential to Editor” section, and submit your "Accept" recommendation.

Reviewer #1: All comments have been addressed

2. Is the manuscript technically sound, and do the data support the conclusions?

Reviewer #1: Yes

3. Has the statistical analysis been performed appropriately and rigorously? 

Reviewer #1: Yes

4. Have the authors made all data underlying the findings in their manuscript fully available?

Reviewer #1: Yes

5. Is the manuscript presented in an intelligible fashion and written in standard English?

Reviewer #1: Yes

6. Review Comments to the Author

Reviewer #1: The author(s) have revised the manuscript in line the earlier issues raised in my review. The manuscript is now more robust.

7. PLOS authors have the option to publish the peer review history of their article (what does this mean?). If published, this will include your full peer review and any attached files.

Reviewer #1: No

---

## [Editor Report · Acceptance letter]

26 Aug 2021

PONE-D-21-06116R1 

Determinants of Birth Registration in India: Evidence from NFHS 2015-16 

Dear Dr. Kumar:

I'm pleased to inform you that your manuscript has been deemed suitable for publication in PLOS ONE. Congratulations! Your manuscript is now with our production department. 

Kind regards, 

on behalf of

Prof. Kannan Navaneetham 

Academic Editor

PLOS ONE